# Characterisation of Pineapple Cultivars under Different Storage Conditions Using Infrared Thermal Imaging Coupled with Machine Learning Algorithms

Maimunah Mohd Ali [1] , Norhashila Hashim [1,2,*] , Samsuzana Abd Aziz [1,2] and Ola Lasekan [3]

1 Department of Biological and Agricultural Engineering, Faculty of Engineering, Universiti Putra Malaysia, Serdang 43400, Selangor, Malaysia; maimunah_mohdali@ymail.com (M.M.A.); samsuzana@upm.edu.my (S.A.A.)
2 SMART Farming Technology Research Centre (SFTRC), Faculty of Engineering, Universiti Putra Malaysia, Serdang 43400, Selangor, Malaysia
3 Department of Food Technology, Faculty of Food Science and Technology, Universiti Putra Malaysia, Serdang 43400, Selangor, Malaysia; lasekan@upm.edu.my
* Correspondence: norhashila@upm.edu.my; Tel.: +60-3-97694336; Fax: +60-3-89466425

**Abstract:** The non-invasive ability of infrared thermal imaging has gained interest in various food classification and recognition tasks. In this work, infrared thermal imaging was used to distinguish different pineapple cultivars, i.e., MD2, Morris, and Josapine, which were subjected to different storage temperatures, i.e., 5, 10, and 25 °C and a relative humidity of 85% to 90%. A total of 14 features from the thermal images were obtained to determine the variation in terms of image parameters among the different pineapple cultivars. Principal component analysis was applied for feature reduction in order to prevent any effect of significant difference between the selected features. Several types of machine learning algorithms were compared, including linear discriminant analysis, quadratic discriminant analysis, support vector machine, k-nearest neighbour, decision tree, and naïve Bayes, to obtain the best performance for the classification of pineapple cultivars. The results showed that support vector machine achieved the best performance from the combination of optimal image parameters with the highest classification rate of 100%. The ability of infrared thermal imaging coupled with machine learning approaches can be potentially used to distinguish pineapple cultivars, which could enhance the grading and sorting processes of the fruit.

**Keywords:** storage temperature; thermal imaging; machine learning; pineapple; cultivar classification

## 1. Introduction

Pineapple belongs to the *Ananas* genus of the Bromeliaceae family, which has been cultivated commercially in subtropical and tropical regions worldwide [1,2]. The appealing aroma and abundant nutritional composition of pineapple make it highly favoured by consumers. Pineapples can be eaten fresh, dried, or processed in various products such as jam, juice, pickle, candy, canned syrup, and beverages. The fruit also contains bromelain, which acts as an enzyme to break down protein and serves as a good source for various health benefits [3]. In terms of pineapple cultivation grown worldwide, the fruit cultivars are classified into four main groups: Queen, Smooth Cayenne, Red Spanish, and Pernambuco [4]. Generally, the pineapple cultivars are distinguishable by fruit weight, shape, size, colour, bioactive compounds, and physiochemical composition depending on the fruit characteristics. Due to the unique criteria of pineapple, it is necessary to differentiate the differences between the fruit cultivars to match the preferences of the consumer.

The pineapple fruit from different cultivars has been classified using various analytical methods such as the evaluation of bioactive compounds [5], determination of physicochemical properties [6,7], carotenoid detection using high-performance liquid chromatography [8], and volatile fingerprinting [9]. However, these methods are labour- and

time-intensive due to the complex analysis and specialised skills required. In particular, conventional methods are heavily influenced by human labour, which is very subjective and requires extensive operation [10,11]. The market demand for different types of pineapple cultivars is not only associated with the external quality of the fruit, but also the internal quality, which is prone to defects, specifically during storage. Storage is a beneficial factor in the postharvest chain of pineapple since the fruit availability and quality need to be monitored before distributing to the commercial market. For this reason, the conventional methods, which are destructive in nature for assessing fruit quality during storage, remain a huge challenge. Thus, a reliable and non-destructive technique for classifying pineapple cultivars is required to obtain efficient and robust results.

In recent years, infrared thermal imaging has been introduced as a reliable and non-destructive evaluation technique for monitoring the quality and safety of various agricultural products. Infrared thermal imaging is a non-contact technique that converts the temperature pattern of a material into visible images for the analysis of feature extraction [12,13]. The applications of infrared thermal imaging have gained much interest in the fruit industry due to the cost reduction in operating devices, rapid measurement, and simple procedure in obtaining data regarding the material [14]. Furthermore, Hussain et al. [15] described that the ability of monitoring temperature in food processing required no external source of energy for imaging. To date, various researchers have widely investigated the potential of the infrared thermal imaging technique for the quality inspection of fruit. The previous studies involving the applications of infrared thermal imaging include immature citrus fruit detection [16], bruising classification of pears [17], fungal infection of potato tubers [18], maturity grading of mangoes [19], temperature estimation of apples [20], chilling injury of guavas [21], and the disease detection of tomatoes [22].

Nowadays, various machine learning methods have been developed to quantify the quality and safety evaluation of different kinds of fruit. In this sense, the integration of infrared thermal imaging coupled with machine learning approaches is considered efficient since the multivariate nature of the algorithm is easy to analyse and produces rapid results. The trend of using machine learning is explored by employing various algorithms such as partial least squares (PLS), support vector machine (SVM), principal component analysis (PCA), random forest, ordinary least squares, stepwise linear regression, and k-nearest neighbour (kNN) [23,24]. While a significant effort has been exerted in investigating the chemical and physical attributes of pineapples, only limited studies have been undertaken in developing predictive and classification systems based on various storage conditions for the fruit. In practical applications, any machine learning classifier can be implemented in such a way that the feature extraction may provide distinct classification rates and increase the model accuracy [25]. Hence, this work attempts to classify pineapple cultivars under different storage conditions using infrared thermal imaging coupled with the machine learning approach. The thermal imaging system extracted image parameters to classify the pineapple cultivars in relation to the different storage conditions using six prominent machine learning algorithms including linear discriminant analysis (LDA), quadratic discriminant analysis (QDA), SVM, kNN, decision tree, and naïve Bayes. The specific objectives of this research were: (1) to determine the image parameters among different pineapple cultivars under different storage conditions, and (2) to compare the performance metrics of machine learning algorithms based on the image parameter features.

## 2. Materials and Methods

### 2.1. Fruit Samples

Three different pineapple cultivars, MD2, Morris, and Josapine, were harvested at a ripening stage of Index 2 (50% unripe, glossy dark green in colour, with traces of yellow between eyes at the base) from a local farm in Simpang Renggam, Johor, Malaysia. All of the pineapple cultivars were transported immediately to the Biomaterials Processing Laboratory, Universiti Putra Malaysia, after harvest. The fruit samples were stored at three different temperatures: 5 °C (cold storage room), 10 °C (controlled refrigerator), and 25 °C

(air-ventilated laboratory room) and a relative humidity storage environment of 85% to 90%. The fruit samples were randomly numbered without cleaning or treatment prior to the storage to prevent any losses. Thirty pineapple samples were randomly selected into four interval groups (Day 0, Day 7, Day 14, and Day 21) for each cultivar. The fruits were kept in a laboratory room condition (25.0 ± 1.0 °C, 90.0 ± 0.5% RH) before starting the sample preparation procedure. To determine the pineapple classification, a total of 1080 samples (360 of each cultivar) were randomly selected. All of the fruit samples from the three different cultivars were analysed and divided into training and testing datasets. Figure 1 shows the images of different varieties of pineapples stored at the three storage temperatures. Based on the random classification algorithms, 756 pineapple samples were used in the training dataset, whereas the remaining 324 samples were chosen for the testing dataset. The fruit samples in the training and testing datasets remained the same for all of the algorithms in order to compare the performance of different models.

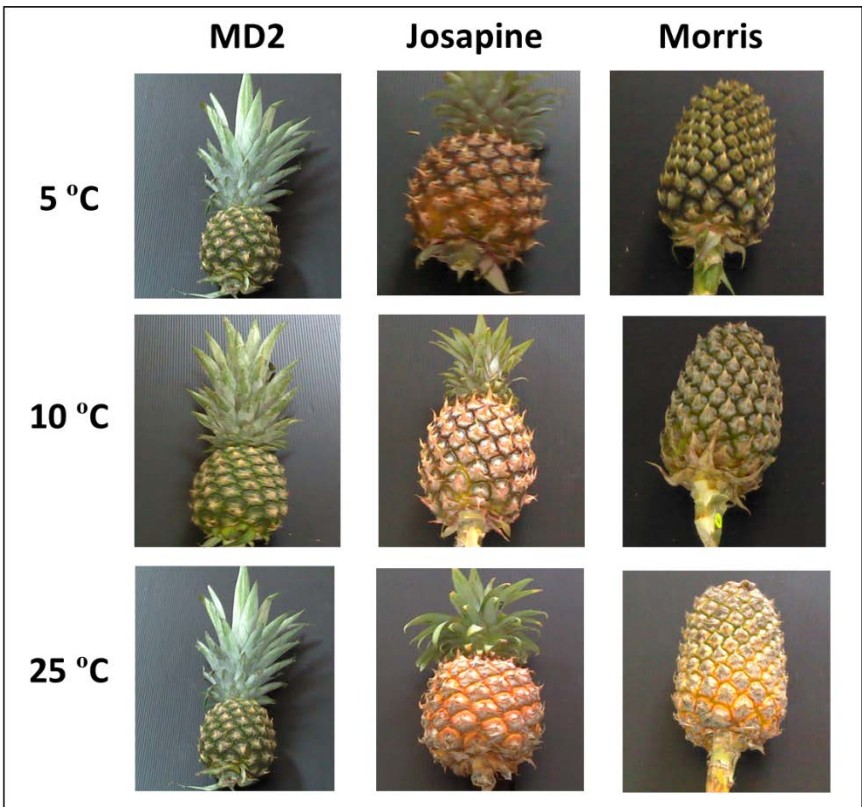

**Figure 1.** Images of different varieties of pineapples stored at three storage temperatures.

### 2.2. Infrared Thermal Imaging

An infrared thermal imaging system consisting of a thermographic camera (FLIR E60, FLIR Systems, King Hills, United Kingdom), sample holder, and a computer equipped with processing software was developed. The thermographic camera with temperature control in the range of −20 °C to +650 °C was equipped with 0.7 to 1.4 μm, an infrared resolution of 320 × 240 pixels, and a thermal sensitivity less than 0.05 °C. A lens with a field of view of 25° × 19° and five measurement modes was used with the thermographic camera. The distance between the camera lens and the fruit surface was set to 0.4 m to capture the thermal images. The image acquisition of the fruits was performed immediately upon removing the samples from the storage at ambient temperature for the identification of pineapple cultivars under different storage conditions. The thermal images were acquired at a room temperature of 25 °C to avoid potential fluctuations in temperatures of the thermal camera due to continuous operation. A total of 3240 thermal images were obtained for the overall fruit samples.

### 2.3. Thermal Image Processing

Feature extraction was carried out to select the region of interest (ROI) of the thermal images. Prior to feature extraction, the image processing and segmentation steps were performed to facilitate the cultivar classification of pineapples based on the selected image features. The image processing and segmentation steps of the thermal image are described in Figure 2. The image processing steps comprised the removal of image shadow, background noise elimination, and the separation of the ROI from the image background. The thermal image was converted to a greyscale image to facilitate the feature extraction. The Otsu thresholding technique was performed to obtain the threshold level in order to convert the greyscale image to a binary image. In this case, the image segmentation could be maximised by dividing the image into the background and selected ROI. The shape and pixel value features were obtained by feature extraction using MATLAB Version R2020a software (The MathWorks, Natick, MA, USA). A total of 14 image features, accumulated from six pixel values (maximum intensity, mean intensity, minimum intensity, maximum of ROI, mean of ROI, and minimum of ROI) and eight shapes (centroid, area, eccentricity, perimeter, orientation, major axis length, minor axis length, and extent) features were selected for each pineapple cultivar. The respective values in all selected features were described in the pixel count, which was stored as the classification variables.

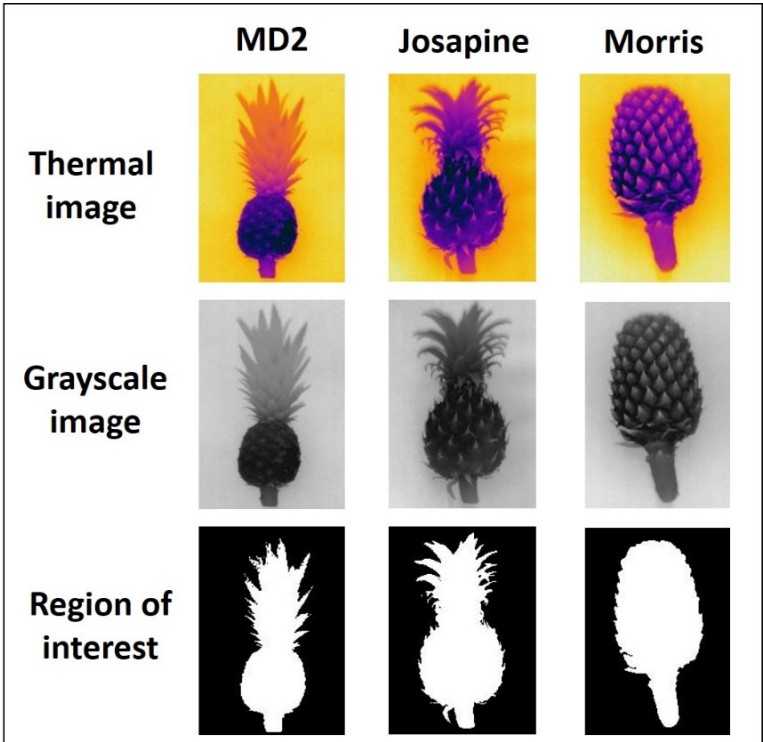

**Figure 2.** Thermal image analysis using image segmentation processes.

### 2.4. Variable Selection Using Principal Component Analysis

PCA was applied to obtain feature extraction according to the high component loading of the principal components (PCs). The PCA was carried out using the Unscrambler X Version 10.3 (CAMO Software, Oslo, Norway). The whole dataset was randomly split into calibration (70%) and cross-validation (30%) sets, respectively. The full cross-validation method was applied to the PCA. Subsequently, the PCA model was applied to the validation set to evaluate the classification ability of the model. The relationship between the storage conditions was highlighted, which was associated with the selected features based on the thermal images of the pineapple. The variables were selected corresponding to the maximum eigenvalues in order to visualise the distribution of the pineapple cultivars. Two PCs including PC1 and PC2 were used from the largest contribution of total variance to

demonstrate the variation of image parameters of the pineapple images. The proportion of total variability and the eigenvalues were determined based on the PCA. In this study, PCA score plots and correlation loading were obtained to choose the optimal image parameters for the cultivar classification of pineapples.

*2.5. Machine Learning Algorithms*

Six different machine learning algorithms were studied to classify the pineapple cultivars in relation to different storage conditions based on the image parameters including LDA, QDA, SVM, kNN, decision tree, and naïve Bayes. All of the machine learning algorithms were built using MATLAB Version R2020a software (The MathWorks, USA) in order to discriminate the pineapple cultivars. The flowchart for the classification of pineapple cultivars using infrared thermal imaging is illustrated in Figure 3.

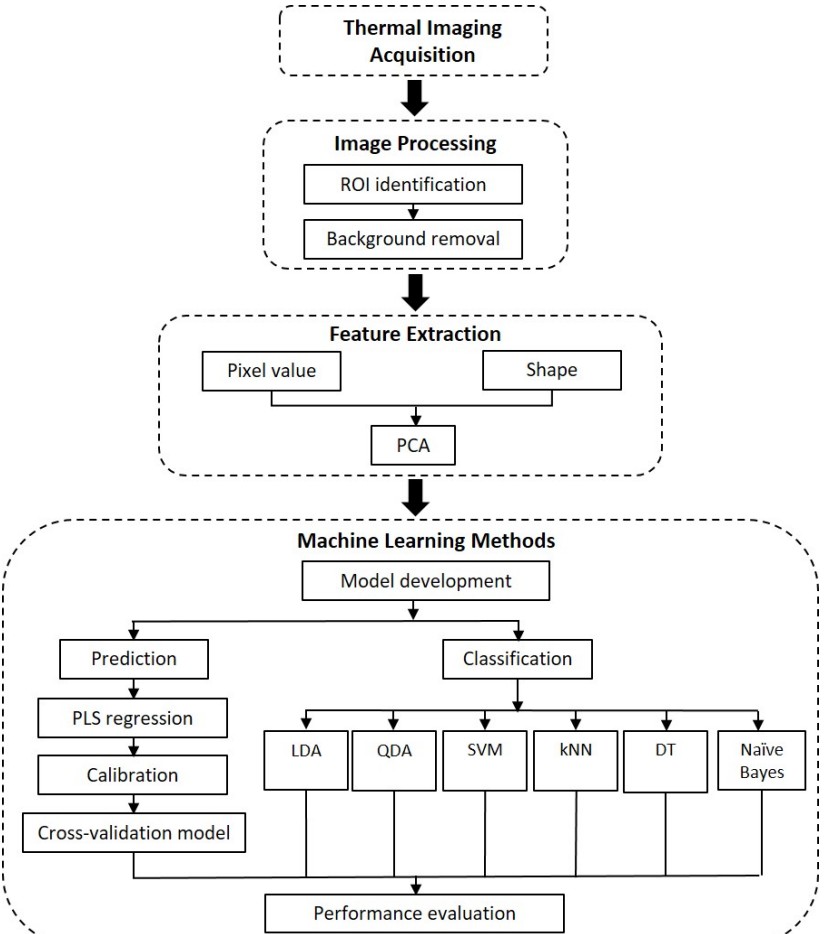

**Figure 3.** Flowchart for the classification of pineapple cultivars using infrared thermal imaging technique.

2.5.1. Linear Discriminant Analysis (LDA)

The LDA algorithm applies a linear transformation to obtain the directions of maximum variance of input data [25]. The LDA is known as a supervised method that is widely used to classify objects for data generation and classification tasks. This algorithm aims to maximise interclass variability by employing several groupings for the classification model. In the present study, the LDA method was applied to develop classification models for the discrimination of pineapple cultivars according to the different storage conditions.

### 2.5.2. Quadratic Discriminant Analysis (QDA)

Another discriminant analysis that is used for the classification approach is the QDA method. The QDA method is based on a quadratic model that generates the classification task from the testing data. Apart from that, QDA allows the covariance of each class instead of pooling the whole sample [26]. To run both LDA and QDA algorithms, 70% of the overall datasets were considered as training data and the remaining 30% as the testing data.

### 2.5.3. Support Vector Machine (SVM)

The SVM is a supervised learning method specifically to evaluate data for regression and classification problems. This approach uses a hyperplane that has the largest distance to the nearest training data from any class to achieve good classification [27]. The SVM can also discriminate non-linear data using a kernel function *k (x, y)* according to the related task. Further, SVM has been used for binary classification in order to achieve accurate results with a small amount of data sampling. Using the SVM method, 30% was considered as testing data and the rest was selected as training data. To establish a computational load, several common kernels can be used such as linear, sigmoid, radial basis, and polynomial functions [28]. For this purpose, the radial basis function with the penalty coefficient value ($\gamma$) at 1 was selected for the SVM model. The radial basis function is defined in Equation (1):

$$K\ (x,\ xi) = \exp(-\gamma||x - xi||^2),\ \ \gamma > 0 \tag{1}$$

### 2.5.4. K-Nearest Neighbour (kNN)

The kNN is a supervised method particularly for regression and classification analyses in which K denotes the number of neighbours [29]. The input data comprises the k-closest training data from the features whereas the output data is the classified number of instances. Generally, the distances between the training and testing data were calculated to obtain the k-nearest neighbour decision factor [30]. In order to categorise the testing data from several classes, the minimum distance based on the training data was determined. Initially, one and the highest K values were evaluated to obtain the optimum number of neighbours. In this study, the number of neighbours obtained to develop the classification model was equalled to 10.

### 2.5.5. Decision Tree

Decision tree is a decision support algorithm that implements a tree-like model to describe a possible result as a function of independent variables. This approach is widely used in classification models due to the easy interpretation and good reliability with the database systems [31]. The tree model was developed by repetitive splits of subsets based on the training datasets. Typically, each split was described by a simple rule according to the single independent variable. An optimal feature was chosen as the basis for the division set in order to construct the tree model. The training datasets were also randomly divided into subsets to obtain the best classification results. The tree model was developed when all of the subsets fitted to the leaf nodes. A Gini index was used to choose a split for the benchmark of partition in the decision tree. The process was repeated until the tree model had a maximum size once the split was determined. For this study, the maximum number for decision trees was 16 for the classification model.

### 2.5.6. Naïve Bayes

Naïve Bayes is a parametric and supervised technique according to the Bayes' theorem along with strong independence associations between the data features [32]. The preceding probability of classes was determined using the class relative frequency distribution. Naïve Bayes allows a normal distribution between classes by calculating the standard deviation and average of the training dataset via maximum likelihood estimation [33]. In this study, a classification model from the naïve Bayes was applied to testing data based on the largest posterior probability.

*2.6. Data Analysis*

Significant differences of image parameters at different cultivars were identified using analysis of variance (ANOVA). The mean comparison was determined by Tukey's test based on $p < 0.05$ using the SAS software (Version 9.4, SAS Institute, Cary, NC, USA).

The performance of the machine learning algorithms was evaluated in terms of classification accuracy (%). The classification methods were carried out using tenfold cross-validation using the selected image parameters from the feature extraction. The mean accuracy was obtained for each classification trial in order to compare the performance among the machine learning algorithms. Generally, a classification model was evaluated based on the high accuracy rate from the classification trials. Further, a confusion matrix was used to describe the estimation rate of the machine learning algorithm with several variables known as true negatives, true positives, false negatives, and false positives [34].

## 3. Results and Discussion

*3.1. Feature Selection*

The average values of the image parameters from different pineapple cultivars are tabulated in Table 1. All image parameters of different pineapple cultivars had significant differences at the 95% confidence level ($p < 0.05$). The values in all image parameters were calculated as a pixel count. The highest eccentricity and perimeter were found in the MD2 cultivar with the values of 0.72 and 1464.30, respectively. For the Josapine cultivar, the highest values of the image parameters were obtained in area (63976.00), orientation (0.70), and extent (0.84). On the other hand, the image parameter values remained unchanged for minimum intensity (0.54), maximum of ROI (0.97), and minimum of ROI (0.55) for all pineapple cultivars, respectively. As for the remaining image parameters, the highest values were found in Morris including centroid (157.39), major axis length (393.74), minor axis length (292.21), maximum intensity (0.96), mean intensity (0.67), and mean of ROI (0.81). For this reason, the utilisation of image parameters was best described to define the behaviour of the thermal images, contributing to the high dependency based on different pineapple cultivars.

**Table 1.** Average values of image parameters from different pineapple cultivars.

| Image Parameter | Cultivar | | |
|---|---|---|---|
| | MD2 | Josapine | Morris |
| Centroid | 150.49 ± 13.64 [a] | 156.26 ± 4.45 [a] | 157.39 ± 2.69 [a] |
| Area | 57,189.00 ± 13,314.00 [b] | 63,976.00 ± 4481.00 [a] | 63,167.00 ± 3630.00 [a] |
| Eccentricity | 0.72 ± 0.07 [a] | 0.67 ± 0.02 [b] | 0.67 ± 0.02 [b] |
| Perimeter | 1464.30 ± 507.30 [a] | 1437.00 ± 332.00 [a] | 1226.90 ± 297.40 [a] |
| Orientation | −3.81 ± 35.97 [b] | 0.70 ± 11.42 [a] | −0.56 ± 1.41 [ab] |
| Major axis length | 372.93 ± 45.01 [b] | 389.62 ± 13.92 [a] | 393.74 ± 10.93 [a] |
| Minor axis length | 260.08 ± 54.25 [b] | 288.03 ± 13.71 [a] | 292.21 ± 6.80 [a] |
| Extent | 0.82 ± 0.07 [b] | 0.84 ± 0.04 [a] | 0.82 ± 0.05 [b] |
| Maximum intensity | 0.94 ± 0.06 [b] | 0.95 ± 0.05 [a] | 0.96 ± 0.04 [a] |
| Mean intensity | 0.64 ± 0.05 [b] | 0.62 ± 0.33 [b] | 0.67 ± 0.06 [a] |
| Minimum intensity | 0.54 ± 0.05 [a] | 0.54 ± 0.04 [a] | 0.54 ± 0.05 [a] |
| Maximum of ROI | 0.97 ± 0.01 [a] | 0.97 ± 0.01 [a] | 0.97 ± 0.01 [a] |
| Mean of ROI | 0.77 ± 0.06 [c] | 0.79 ± 0.06 [b] | 0.81 ± 0.05 [a] |
| Minimum of ROI | 0.55 ± 0.05 [b] | 0.55 ± 0.04 [ab] | 0.55 ± 0.08 [a] |

Values are the mean ± standard deviation. Different letters in the same row indicate significant differences ($p < 0.05$). ROI refers to region of interest.

In order to explore the dataset, a quantitative feature comparison was determined to evaluate the differences between the pineapple cultivars for the classification task. It was revealed that the distribution of image parameter values was significantly different between all pineapple cultivars. Considering the difference in the fruit cultivar, the temperature differences were attributed to the selected features of the thermal images [35]. The changes

of image parameters showed the pixel distribution based on the temperature mapping attained at the surface of the pineapples for different fruit cultivars. The image features were the basic elements for the cultivar discrimination, which would be useful in determining the characteristics and parameters of the sample [36]. Apart from that, the output from the feature selection of the image parameters is applied as input for developing machine learning algorithms to further improve the classification accuracy.

### 3.2. Relationship Analysis

The image parameters derived from the pixel values and shape features were used to distinguish the pineapple cultivars. The linear correlation coefficients between all image parameters of pineapple images are shown in Figure 4. Among all of the image parameters, minimum intensity was highly correlated with eccentricity with a correlation coefficient (r) of 0.98. In contrast, extent was negatively correlated (r = −0.97) with minimum intensity. A low correlation was found between perimeter and major axis length (r = 0.56). It was demonstrated that the centroid was positively correlated with maximum intensity, area, extent, and orientation with linear correlation coefficients ranging from 0.68 to 0.94. Based on the pixel value features, only the maximum intensity was found to be positively correlated with all of the shape features.

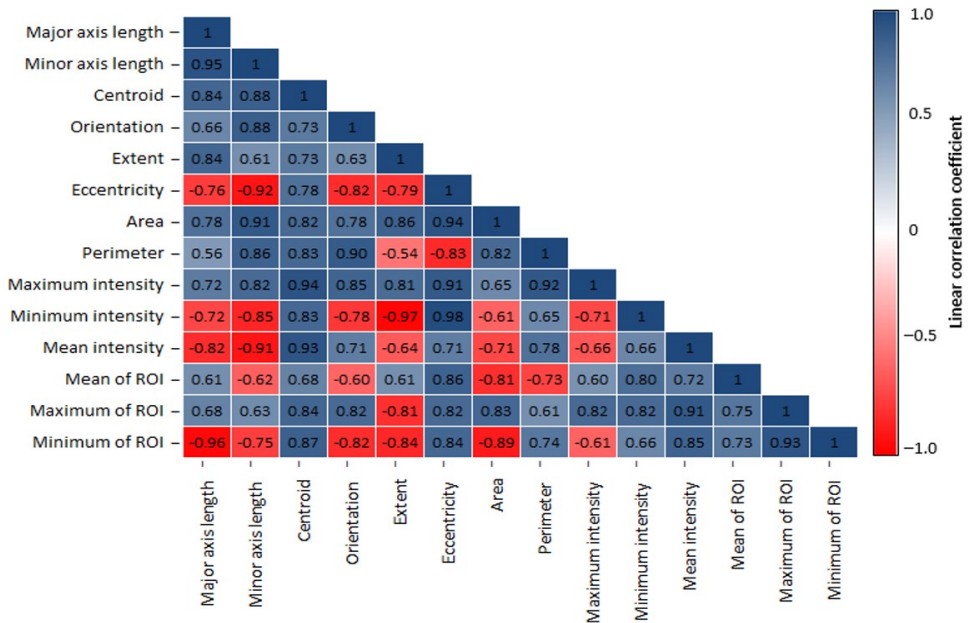

**Figure 4.** Linear correlation coefficients between all image parameters of pineapple images.

In addition, specific image parameters with high correlations could be chosen to be associated with a certain feature for the classification. A high correlation was observed due to the variation between the fruit cultivars indicating the relationship between the pixel values and shape features of the pineapples. Koklu and Ozkan [28] identified different types of dry beans using shape and dimensional features taken from two-dimensional images for classifying the varieties. Feature extraction was generated to achieve feature values, which were used to statistically compare between the classes for the classification [37]. In this case, linear correlation has been used to investigate the relationships among fruit properties and cultivars as well as to obtain discriminatory features [32]. In relation to the relationship analysis, all of the image parameters were significantly correlated, which were feasible to determine the classification of pineapple cultivars according to different storage conditions.

### 3.3. Classification Results Using PCA

Based on the image parameters of pineapple images, the effectiveness of PCA models was evaluated as shown in Figure 5. The PCA model was established to verify the

clustering ability of the three different pineapple cultivars, namely MD2, Josapine, and Morris. It was observed that the three different pineapple cultivars were successfully classified by two PCs with PC1 (97%) and PC2 (3%), accumulating a total variance of 100%, respectively (Figure 5a). The classification results using PCA models were in agreement with Kuzy et al. [38] who demonstrated high capability in terms of clustering patterns between Farthing and Meadowlark berries. Further, the findings revealed that the three pineapple cultivars showed positive scores along both PC1 and PC2 according to the variability loadings.

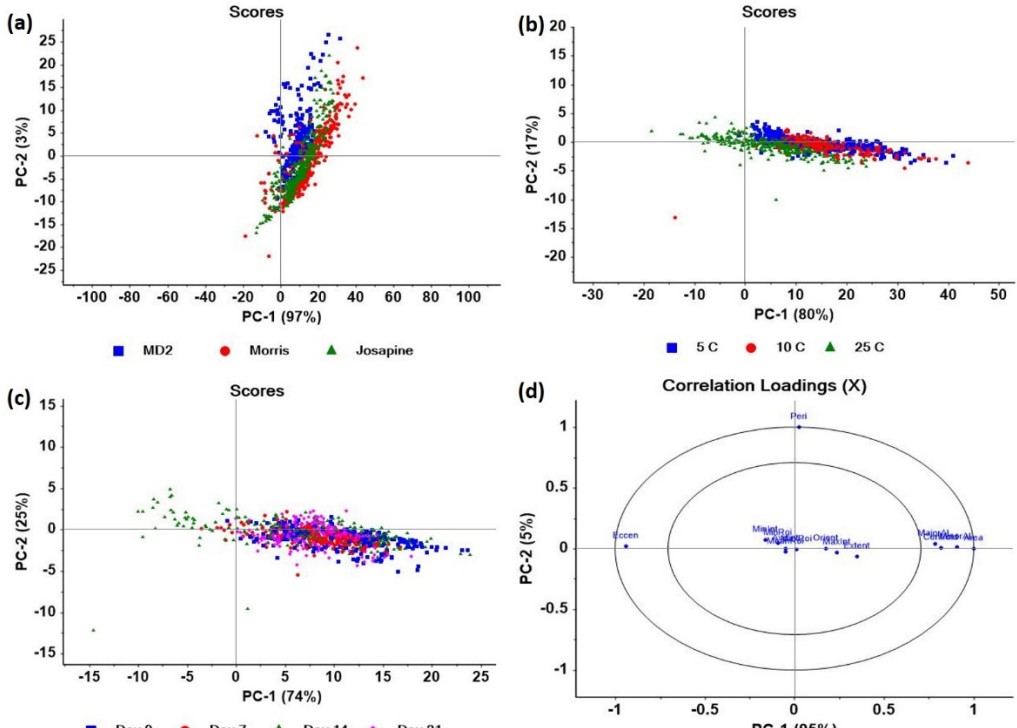

**Figure 5.** Principal component analysis score plots showing the variation of image parameters based on: (**a**) fruit cultivars, (**b**) storage temperatures, (**c**) storage days, and (**d**) correlation loading plot of image parameters.

According to the clustering performance based on the storage temperatures, the results clearly distinguished the variations by two components of PC1 (80%) and PC2 (17%) with total variances of 97% (Figure 5b), respectively. For this reason, it could be explained that each pineapple cultivar subjected to three different storage temperatures (5, 10, and 25 °C) showed significant variations in the quality attributes of the fruit. As a result, all of the pineapple cultivars stored at three different storage temperatures were correctly discriminated according to the variability of the image parameters. Additionally, the findings successfully discriminated the variations of image parameters in relation to different storage days as proportioned by PC1 (74%) and PC2 (25%), resulting in a total variance of 99% (Figure 5c). In order to investigate the effect of the image parameters, both PC1 and PC2 signified the ability of the infrared thermal imaging technique to distinguish the variations observed in the pineapple samples during storage.

With respect to the classification scores corresponding with the selected image parameters and different pineapple cultivars, the correlation loadings were strongly correlated with PC1 (95%) and PC2 (5%), accumulating a total variance of 100% (Figure 5d). The results indicated that maximum intensity, mean intensity, minimum intensity, maximum of ROI, mean of ROI, and minimum of ROI, orientation, and extent described the best combination of image parameters for the classification of pineapple cultivars were subjected to the interior ellipse in the PCA plot. Furthermore, the loading scores aided in the detection

of optimal image parameters, which were suitable for the classification task of pineapple cultivars based on different storage conditions. All pineapple samples consisting of MD2, Josapine, and Morris were correctly distinguished in their respective clusters according to their cultivar-related functions. The discrimination of pineapple cultivars based on the image parameters was important as an indicator to provide a clear visualisation influenced by the different storage conditions. These observations were similar to those of Sanchez et al. [39], who reported total variances of 100% for the classification of sweet potato varieties based on the quality properties during storage. With regard to the experimental factors used, the PCA method required at least two variables to evaluate the classification performance of the samples [40]. Thus, the baseline data could be applied to evaluate the variability of other physicochemical properties of pineapples for a wide range of cultivars and experimental factors.

*3.4. Comparison of Machine Learning Models*

The implementation of machine learning algorithms was developed to determine the classification accuracy for the detection of pineapple cultivars based on different storage conditions using an infrared thermal imaging technique. The classification performance of the pineapple cultivars at different storage days and temperatures using the LDA method is presented in Table 2. The LDA results were obtained according to the classification performance of pineapple cultivars at 25 °C (93.21–98.03%), followed by 10 °C (92.49–97.91%), and 5 °C (92.81–97.64%), respectively. It can be denoted that the classification accuracy of the LDA models increased over storage days for all pineapple cultivars at different storage temperatures. The LDA models attained the highest classification accuracies recorded at 25 °C for both Day 0 (94.67%) and Day 7 (96.39%) from the Josapine cultivar, respectively. The Morris cultivar obtained the highest classification accuracy among all storage days at 25 °C (98.03%) for Day 21. The performance of the infrared thermal imaging technique based on LDA was found to be feasible, which obtained overall classification rates up to 96.25% under different storage conditions for all pineapple cultivars.

**Table 2.** Classification performance of pineapple cultivars at different storage days and temperatures using linear discriminant analysis.

| Cultivar | Temperature | Classification Accuracy (%) | | | | Overall Classification Rate (%) |
|---|---|---|---|---|---|---|
| | | **Day 0** | **Day 7** | **Day 14** | **Day 21** | |
| MD2 | 5 °C | 92.81 | 93.84 | 93.09 | 95.82 | 93.89 |
| | 10 °C | 93.02 | 94.16 | 94.74 | 96.18 | 94.53 |
| | 25 °C | 94.60 | 94.98 | 96.92 | 97.24 | 95.94 |
| Josapine | 5 °C | 93.95 | 95.83 | 97.22 | 97.64 | 96.16 |
| | 10 °C | 92.49 | 94.70 | 95.83 | 97.91 | 95.23 |
| | 25 °C | 94.67 | 96.39 | 96.84 | 97.11 | 96.25 |
| Morris | 5 °C | 93.05 | 94.21 | 95.09 | 97.26 | 94.90 |
| | 10 °C | 92.94 | 94.78 | 96.31 | 94.29 | 94.58 |
| | 25 °C | 93.21 | 94.87 | 95.22 | 98.03 | 95.33 |

The classification performance of the pineapple cultivars at different storage days and temperatures using the QDA method is shown in Table 3. The findings were described based on the classification performance of the pineapple cultivars at 25 °C (92.66–99.28%), followed by 10 °C (92.53–98.47%) and 5 °C (93.85–97.60%), respectively. The classification accuracy of the QDA models gradually increased over the storage days for all pineapple cultivars at different storage temperatures. The QDA models obtained the highest classification accuracies recorded at 25 °C for both Day 7 (95.71%) and Day 21 (99.28%) from the Josapine cultivar, respectively. Based on the QDA results, it was signified that the overall classification rates achieved up to 96.40% under different storage conditions for all pineapple cultivars.

**Table 3.** Classification performance of pineapple cultivars at different storage days and temperatures using quadratic discriminant analysis.

| Cultivar | Temperature | Classification Accuracy (%) | | | | Overall Classification Rate (%) |
|---|---|---|---|---|---|---|
| | | Day 0 | Day 7 | Day 14 | Day 21 | |
| MD2 | 5 °C | 94.69 | 95.32 | 95.99 | 97.20 | 95.80 |
| | 10 °C | 92.53 | 95.37 | 96.04 | 97.45 | 95.35 |
| | 25 °C | 94.88 | 95.21 | 96.56 | 98.93 | 96.40 |
| Josapine | 5 °C | 93.85 | 94.74 | 95.88 | 97.15 | 95.41 |
| | 10 °C | 94.60 | 95.19 | 96.33 | 98.47 | 96.15 |
| | 25 °C | 93.09 | 95.71 | 96.95 | 99.28 | 96.26 |
| Morris | 5 °C | 94.81 | 95.44 | 96.28 | 97.60 | 96.03 |
| | 10 °C | 93.95 | 94.02 | 95.07 | 97.26 | 95.08 |
| | 25 °C | 92.66 | 94.86 | 97.57 | 98.45 | 95.89 |

The classification performance of pineapple cultivars at different storage days and temperatures using the SVM method is demonstrated in Table 4. The findings were evaluated according to the classification performance of the pineapple cultivars at 25 °C (96.32–99.93%), followed by 10 °C (94.96–99.72%), and 5 °C (96.02–99.62%), respectively. It was also observed that the classification accuracy of the SVM models increased over storage days for all pineapple cultivars at different storage temperatures. The SVM models achieved the highest classification accuracies recorded at 25 °C for Day 7 (99.11%), Day 14 (99.92%), and Day 21 (99.93%) from the Morris cultivar, respectively. Similarly, the Morris cultivar obtained the highest classification accuracy for Day 0 (98.26%) which was recorded at 5 °C. Moreover, it was revealed that the overall classification rates achieved up to 99.30% under different storage conditions for all pineapple cultivars.

**Table 4.** Classification performance of pineapple cultivars at different storage days and temperatures using support vector machine.

| Cultivar | Temperature | Classification Accuracy (%) | | | | Overall Classification Rate (%) |
|---|---|---|---|---|---|---|
| | | Day 0 | Day 7 | Day 14 | Day 21 | |
| MD2 | 5 °C | 96.42 | 98.01 | 99.29 | 98.98 | 98.18 |
| | 10 °C | 96.99 | 98.76 | 98.83 | 99.29 | 98.47 |
| | 25 °C | 97.18 | 98.34 | 98.36 | 99.36 | 98.31 |
| Josapine | 5 °C | 96.02 | 97.86 | 98.79 | 99.62 | 98.07 |
| | 10 °C | 97.28 | 97.96 | 98.38 | 99.34 | 98.24 |
| | 25 °C | 96.32 | 97.74 | 98.96 | 99.29 | 98.08 |
| Morris | 5 °C | 98.26 | 98.13 | 98.90 | 99.47 | 98.69 |
| | 10 °C | 94.96 | 98.62 | 98.87 | 99.72 | 98.02 |
| | 25 °C | 98.25 | 99.11 | 99.92 | 99.93 | 99.30 |

The classification performance of pineapple cultivars at different storage days and temperatures using the kNN method is presented in Table 5. The kNN results were obtained according to the classification performance of the pineapple cultivars at 25 °C (95.83–99.93%), followed by 10 °C (96.42–99.75%) and 5 °C (95.39–99.46%), respectively. It was demonstrated that the classification accuracy of the kNN models increased over the storage days for all pineapple cultivars at different storage temperatures. The kNN models obtained the highest classification accuracies recorded at 25 °C for Day 7 (98.41%), Day 14 (99.48%), and Day 21 (99.93%) from the Morris cultivar, respectively. Likewise, the Morris cultivar also attained the highest classification accuracy for Day 0 (97.49%), which was recorded at 10 °C. In addition, the overall classification rates achieved up to 98.70% under different storage conditions for all pineapple cultivars.

**Table 5.** Classification performance of pineapple cultivars at different storage days and temperatures using k-nearest neighbour.

| Cultivar | Temperature | Classification Accuracy (%) | | | | Overall Classification Rate (%) |
|---|---|---|---|---|---|---|
| | | Day 0 | Day 7 | Day 14 | Day 21 | |
| MD2 | 5 °C | 95.39 | 96.08 | 97.33 | 99.34 | 97.04 |
| | 10 °C | 96.42 | 97.32 | 97.20 | 99.28 | 97.56 |
| | 25 °C | 96.75 | 96.34 | 97.37 | 98.46 | 97.23 |
| Josapine | 5 °C | 96.38 | 97.46 | 97.99 | 98.42 | 97.56 |
| | 10 °C | 97.47 | 97.92 | 97.35 | 98.18 | 97.73 |
| | 25 °C | 95.83 | 96.05 | 97.48 | 99.31 | 97.17 |
| Morris | 5 °C | 96.07 | 96.48 | 97.51 | 99.46 | 97.38 |
| | 10 °C | 97.49 | 97.90 | 98.72 | 99.75 | 98.47 |
| | 25 °C | 96.97 | 98.41 | 99.48 | 99.93 | 98.70 |

The classification performance of pineapple cultivars at different storage days and temperatures using the decision tree method is tabulated in Table 6. The findings were achieved based on the classification performance of the pineapple cultivars at 10 °C (96.37–99.95%), followed by 25 °C (94.59–99.86%) and 5 °C (95.20–99.59%), respectively. It was signified that the classification accuracy of the decision tree models significantly increased over the storage days for all pineapple cultivars at different storage temperatures. The decision tree models achieved the highest classification accuracies recorded at 25 °C for Day 7 (99.86%) and Day 14 (99.74%) from the Morris cultivar, respectively. It was also revealed that the overall classification rates achieved up to 98.67% under different storage conditions for all pineapple cultivars.

**Table 6.** Classification performance of pineapple cultivars at different storage days and temperatures using decision tree.

| Cultivar | Temperature | Classification Accuracy (%) | | | | Overall Classification Rate (%) |
|---|---|---|---|---|---|---|
| | | Day 0 | Day 7 | Day 14 | Day 21 | |
| MD2 | 5 °C | 95.22 | 96.36 | 96.89 | 98.31 | 96.70 |
| | 10 °C | 96.37 | 97.89 | 98.10 | 99.58 | 97.99 |
| | 25 °C | 97.59 | 97.97 | 99.71 | 99.42 | 98.67 |
| Josapine | 5 °C | 96.35 | 97.04 | 97.89 | 99.43 | 97.68 |
| | 10 °C | 97.36 | 98.48 | 98.94 | 99.42 | 98.55 |
| | 25 °C | 97.58 | 98.38 | 98.36 | 99.53 | 98.46 |
| Morris | 5 °C | 95.20 | 97.24 | 99.59 | 99.21 | 97.81 |
| | 10 °C | 96.57 | 98.48 | 99.27 | 99.95 | 98.57 |
| | 25 °C | 94.59 | 99.86 | 99.74 | 99.63 | 98.46 |

The classification performance of the pineapple varieties at different storage days and temperatures using the naïve Bayes method is shown in Table 7. The promising naïve Bayes results were accounted according to the classification performance of the pineapple cultivars at 5 °C (95.27–99.96%), followed by 10 °C (95.09–99.96%), and 25 °C (93.67–99.92%), respectively. Based on the results, the classification accuracy of the naïve Bayes models increased over the storage days for all pineapple varieties at different storage temperatures. The naïve Bayes models obtained the highest classification accuracies recorded at 10 °C for Day 21 (99.96%) from the Morris cultivar. The Josapine cultivar also obtained the highest classification accuracy at 10 °C (97.49%) which was recorded at Day 7. It was also found that the overall classification rates achieved up to 98.03% under different storage conditions for all pineapple cultivars. These findings inferred that the changes in image parameters of pineapple cultivars using the infrared thermal imaging technique could show promising use in monitoring various storage conditions.

**Table 7.** Classification performance of pineapple cultivars at different storage days and temperatures using naïve Bayes.

| Cultivar | Temperature | Classification Accuracy (%) | | | | Overall Classification Rate (%) |
|----------|-------------|-------|-------|--------|--------|---------|
| | | Day 0 | Day 7 | Day 14 | Day 21 | |
| MD2 | 5 °C | 95.27 | 95.69 | 97.58 | 99.14 | 96.92 |
| | 10 °C | 96.58 | 97.47 | 98.51 | 99.55 | 98.03 |
| | 25 °C | 93.67 | 95.56 | 98.15 | 98.56 | 96.49 |
| Josapine | 5 °C | 95.95 | 96.31 | 98.97 | 99.96 | 97.80 |
| | 10 °C | 96.73 | 97.49 | 98.12 | 99.43 | 97.94 |
| | 25 °C | 94.88 | 97.46 | 99.32 | 99.89 | 97.89 |
| Morris | 5 °C | 97.36 | 96.59 | 97.13 | 99.45 | 97.63 |
| | 10 °C | 96.32 | 95.09 | 98.25 | 99.96 | 97.41 |
| | 25 °C | 95.97 | 96.59 | 97.97 | 99.92 | 97.61 |

In general, all of the machine learning algorithms succeeded in achieving up to 99.30% of the overall classification rates in distinguishing pineapple cultivars according to various storage conditions. The typical trend of classification accuracy was enhanced in the large total number of features selected from the feature extraction [38]. Regardless of the discrepancy in the classification accuracies between the pineapple cultivars, it should be noted that the reference measurement described the significant changes in image parameters. Vélez Rivera et al. [41] obtained a success rate of 90% in detecting mechanical defects in mango using several algorithms such as LDA, kNN, and naïve Bayes. In the majority of cases, the high correlation of fruit properties could be predicted based on the selected features from the images [32]. In view of the different storage conditions of the fruit, infrared thermal imaging coupled with machine learning demonstrated strong performance and ability for the given classification applications.

To further classify pineapple cultivars according to the image parameters, selected feature extraction allows the machine learning algorithms to achieve classification accuracy. The comparative performance in terms of classification accuracy for the classification of pineapple cultivars between the machine learning algorithms is monitored based on the optimal combination of image parameters. In this case, the distinct features selected from the image parameters provided a different optimal combination applied for each machine learning algorithm using a confusion matrix. Specifically, eight image parameters were selected including maximum intensity, mean intensity, minimum intensity, maximum of ROI, mean of ROI, minimum of ROI, orientation, and extent based on the feature selection using PCA analysis to achieve the highest performance of classification accuracy. The confusion matrices with average classification rates of different pineapple cultivars using six different machine learning algorithms are illustrated in Figure 6.

It can be demonstrated that the LDA achieved an accuracy of 95%, 94%, and 96% for the correct classification of Josapine, MD2, and Morris, respectively. The highest classification accuracy for correctly classified Josapine (97%), Morris (97%), and MD2 (94%) was achieved by QDA. On the other hand, the SVM outperformed the rest of the machine learning algorithms, with the highest classification rate of 100% for the correct classification of all pineapple cultivars. In the case of the kNN algorithm, both Josapine and MD2 were correctly classified with the highest classification accuracy of 100%. The decision tree reached a good classification accuracy of 98% for Josapine, 95% for MD2, and 99% for Morris, respectively. For the naïve Bayes algorithm, the highest classification accuracy obtained was 98% for the correctly classified MD2 cultivar. The dataset of each pineapple cultivar was validated without retraining the machine learning algorithms in order to test the generalisability to other cultivars. Different algorithms should be employed according to the condition according to the current state of the data analysis in obtaining more accurate classification results.

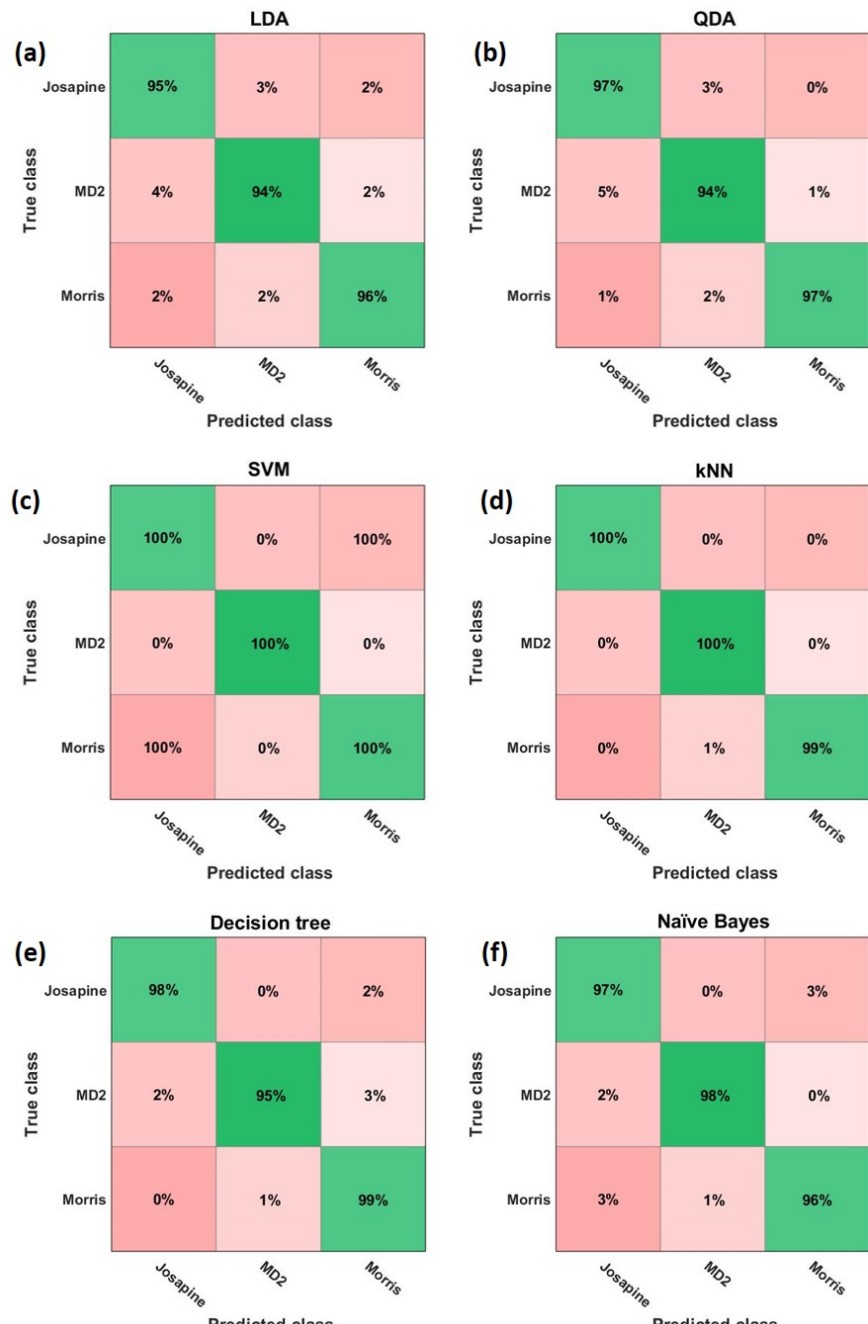

**Figure 6.** Confusion matrices with average classification accuracies of different pineapple cultivars using (**a**) LDA, (**b**) QDA, (**c**) SVM, (**d**) kNN, (**e**) decision tree, and (**f**) naïve Bayes. Green in the confusion matrix describes the correct classification rate, and pink describes the misclassification rate.

With respect to the misclassification of different pineapple cultivars, this could be attributed to the differences in terms of maturity stages and the relationship of variation in quality attributes [42]. In a previous study by van de Looverbosch et al. [37], the superior SVM algorithm was investigated in order to detect two cultivars of pear with several internal disorder severities, which obtained the highest classification accuracy of 95%. Generally, the performance of all of the machine learning algorithms described the highest classification accuracies based on the optimal combination of features of the image parameters. It was observed that all of the machine learning models successfully classified the pineapple cultivars with the highest correct classification up to 100%. Feature extraction may provide the means to choose a minimum number of image parameters for a given

classification task in such a way as to reduce the computational complexity and enhance the model performance [25]. Hence, it can be denoted that all of the machine learning algorithms were able to distinguish between the different pineapples cultivars acquired using the infrared thermal imaging technique.

## 4. Conclusions

The current study evaluated the potential of infrared thermal imaging coupled with machine learning approaches for the cultivar classification of pineapples. The PCA analysis was employed to determine the optimal features to facilitate the cultivar classification of pineapples. By comparing the performance of six different machine learning algorithms, SVM was found to achieve the highest overall classification accuracy of 100%, which could be applied for the discrimination of pineapple cultivars in a non-destructive manner. Additionally, the results demonstrated that feature extraction based on the image parameters allows the machine learning classifiers to obtain high accuracy, which should be considered for the real-time performance of the infrared thermal imaging technique. This evidence provides an insight into the operation involving fruit classification and recognition as an alternative to the manual and tedious conventional methods in order to save an enormous amount of time and effort. Future work may include the application of more sophisticated algorithms such as by employing deep learning for dealing with large datasets. Other algorithms should also be tested to obtain the best combination of feature extraction towards monitoring various fruit classification and recognition as well as other agricultural produce.

**Author Contributions:** Data curation, M.M.A.; investigation, M.M.A.; software, M.M.A.; writing—original draft preparation, M.M.A.; visualization, M.M.A.; supervision, N.H., S.A.A., and O.L.; validation, N.H.; writing—reviewing and editing, N.H.; formal analysis, S.A.A.; conceptualization, O.L. All authors have read and agreed to the published version of the manuscript.

**Funding:** This work was funded by the Putra Grant, GP-IPB (Project code: GP-IPB/2020/9687800).

**Institutional Review Board Statement:** Not applicable.

**Informed Consent Statement:** Not applicable.

**Data Availability Statement:** Data generated during the current study will be made available from the corresponding author on reasonable request.

**Acknowledgments:** The authors are thankful for the support and facilities provided by the Department of Biological and Agricultural Engineering, Faculty of Engineering, Universiti Putra Malaysia.

**Conflicts of Interest:** The authors declare no conflict of interest.

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
