# Peer review of "Characterisation of Pineapple Cultivars under Different Storage Conditions Using Infrared Thermal Imaging Coupled with Machine Learning Algorithms"

_agriculture, doi:10.3390/agriculture12071013_

Round 1

Reviewer 1 Report

Thi paper gives an overall idea of implementation of artificial intelligence in identifying the pineapple cultivars. The experiment is well planned and executed.

o   But the author could have focused on identifying pineapple for major physiological disorders and diseases like internal browning, crown and core rot. If any such data are available?

o   The objective of the study can be emphasized little more in the introduction part

o   What is the RH of the study environment?  Is it 85-90% RH at room condition? Kindly recheck.

o   The details of statistics of PCA studies have not been included in Materials & Methods - Data analysis. Does the PCA and PCA cum ANOVA or PCA alone? Please elaborate on this in the respective section

o   Do you have constructed any biochemical data to be compared with the spectral image of treated pineapples? If so, then that can be given as supplementary data. This will help the readers to break the ambiguity of this paper.

Author Response

Response to Reviewer 1 Comments

Point 1: This paper gives an overall idea of implementation of artificial intelligence in identifying the pineapple cultivars. The experiment is well planned and executed.

But the author could have focused on identifying pineapple for major physiological disorders and diseases like internal browning, crown and core rot. If any such data are available?

Response 1: Thank you for the comment. We agree that physiological disorders and diseases are important aspects, however in this paper we prefer to focus on the fruit cultivar aspect and include your point as a consideration for future study.

Point 2: The objective of the study can be emphasized little more in the introduction part

Response 2: The objective of the study has been revised.

Point 3: What is the RH of the study environment? Is it 85-90% RH at room condition? Kindly recheck.

Response 3: RH of 85-90 % refers to RH storage environment.

Point 4: The details of statistics of PCA studies have not been included in Materials & Methods - Data analysis. Does the PCA and PCA cum ANOVA or PCA alone? Please elaborate on this in the respective section

Response 4: The details of PCA has been added in Materials & Methods-Section 2.4.

Point 5: Do you have constructed any biochemical data to be compared with the spectral image of treated pineapples? If so, then that can be given as supplementary data. This will help the readers to break the ambiguity of this paper.

Response 5: Thank you for this suggestion. It would have been interesting to explore this aspect. However, in our study, this would not be possible because it is beyond the scope of this paper which focus on the utilization of thermal image for the classification of pineapple cultivars. 

Reviewer 2 Report

Title:Characterisation of Pineapple Cultivars under Different Storage Conditions using Infrared Thermal Imaging Coupled with Machine Learning Algorithms

Comments:

The authors studied the characterization of pineapple varieties under different storage conditions by infrared thermography combined with machine learning algorithm. The author first reduced the dimension by principal component analysis, and then compared several machine learning algorithms, such as linear discriminant analysis, quadratic discriminant analysis, support vector machine, k-nearest neighbor, decision tree, Naive Bayes, etc., and obtained the best performance of pineapple variety classification. Many tables are provided to support this study. This study has certain reference significance for the screening and grading of pineapple. However, there are still some problems in the manuscript, which are mainly manifested in the irregular writing and the lack of rigor in some contents. In my opinion, the manuscript requires modifications to meet the standards required by the journal.

I give some general comments justifying the decision and that should be considered:

The key word mentioned fruit quality, but the experiment and discussion in this article did not explore it. Please verify whether this key word is accurate.

Line 99: Why use "i.e."?

The discussion of principal component analysis is not clear and detailed enough.

Line 251: Which variables are called true negative, true positive, false negative, and false positive?

Line 349: Missing period before These.

Please adjust tables format to make them more intuitive.

Line 451: "Notwithstanding" is not accurate, please replace it.

Please check the reference.

There are too few pictures about pineapple samples in the article. Please add more pictures about pineapple sample grading in the article.

Author Response

Response to Reviewer 2 Comments

Point 1: The authors studied the characterization of pineapple varieties under different storage conditions by infrared thermography combined with machine learning algorithm. The author first reduced the dimension by principal component analysis, and then compared several machine learning algorithms, such as linear discriminant analysis, quadratic discriminant analysis, support vector machine, k-nearest neighbor, decision tree, Naive Bayes, etc., and obtained the best performance of pineapple variety classification. Many tables are provided to support this study. This study has certain reference significance for the screening and grading of pineapple. However, there are still some problems in the manuscript, which are mainly manifested in the irregular writing and the lack of rigor in some contents. In my opinion, the manuscript requires modifications to meet the standards required by the journal.

Response 1: Thank you for the comment. We have incorporated all the reviewer’s comments and suggestions to improve the manuscript.

Point 2: The key word mentioned fruit quality, but the experiment and discussion in this article did not explore it. Please verify whether this key word is accurate.

Response 2: The keyword ‘fruit quality’ has been changed to ‘storage temperature’.

Point 3: Line 99: Why use "i.e."?

Response 3: The word ‘i.e.’ has been deleted.

Point 4: The discussion of principal component analysis is not clear and detailed enough.

Response 4: The discussion of principal component analysis has been revised.

Point 5: Line 251: Which variables are called true negative, true positive, false negative, and false positive?

Response 5: True positive is the correct classification rate of pineapple cultivars whereas true negative is the outcome where the model correctly predicts the negative class. False positive is the misclassification rate of pineapple cultivars whereas false negative is the outcome where the model incorrectly predicts the negative class.

Point 6: Line 349: Missing period before “These”.

Response 6: The missing period has been added.

Point 7: Please adjust tables format to make them more intuitive.

Response 7: The tables format has been adjusted.

Point 8: Line 451: "Notwithstanding" is not accurate, please replace it.

Response 8: The word ‘notwithstanding’ has been deleted.

Point 9: Please check the reference.

Response 9: All the references have been thoroughly checked.

Point 10: There are too few pictures about pineapple samples in the article. Please add more pictures about pineapple sample grading in the article.

Response 10: The image of different pineapple varieties stored at different storage temperatures is added (Figure 1).